# Predicting Plant Growth and Development Using Time-Series Images

Chunying Wang [1], Weiting Pan [1], Xubin Song [1], Haixia Yu [2], Junke Zhu [3], Ping Liu [1,*] and Xiang Li [2,*]

1 College of Mechanical and Electronic Engineering, Shandong Agricultural University, Taian 271018, China
2 State Key Laboratory of Crop Biology, College of Life Sciences, Shandong Agricultural University, Taian 271018, China
3 School of Agricultural and Food Engineering, Shandong University of Technology, Zibo 255000, China
* Correspondence: liuping@sdau.edu.cn (P.L.); lixiang@sdau.edu.cn (X.L.)

**Abstract:** Early prediction of the growth and development of plants is important for the intelligent breeding process, yet accurate prediction and simulation of plant phenotypes is difficult. In this work, a prediction model of plant growth and development based on spatiotemporal long short-term memory (ST-LSTM) and memory in memory network (MIM) was proposed to predict the image sequences of future growth and development including plant organs such as ears. A novel dataset of wheat growth and development was also compiled. The performance of the prediction model of plant growth and development was evaluated by calculating structural similarity index measure (SSIM), mean square error (MSE), and peak signal to noise ratio (PSNR) between the predicted and real plant images. Moreover, the optimal number of time steps and the optimal time interval between steps were determined for the proposed model on the wheat growth and development dataset. Under the optimal setting, the SSIM values surpassed 84% for all time steps. The mean of MSE values was 46.11 and the MSE values were below 68 for all time steps. The mean of PSNR values was 30.67. When the number of prediction steps was set to eight, the prediction model had the best prediction performance on the public Panicoid Phenomap-1 dataset. The SSIM values surpassed 78% for all time steps. The mean of MSE values was 77.78 and the MSE values were below 118 for all time steps. The mean of PSNR values was 29.03. The results showed a high degree of similarity between the predicted images and the real images of plant growth and development and verified the validity, reliability, and feasibility of the proposed model. The study shows the potential to provide the plant phenotyping community with an efficient tool that can perform high-throughput phenotyping and predict future plant growth.

**Keywords:** prediction model; machine learning; plant growth and development

## 1. Introduction

Plant growth is a dynamic and complex physiological, biochemical, and metabolic process [1]. Plant phenotyping can assess the complex traits of plant growth and measure individual quantitative parameters [2,3]. These data can be used to quantify small changes in crop growth in a short or long period of time and provide technical support for timely and accurate planting management and plant breeding. Plants grow slowly under natural conditions, which limits experimental cycles. Predicting plant growth and development early and measuring phenotypic traits holds the potential to speed up experimental plant cycles and accelerate plant breeding by reducing the time of plant growth, imaging, and measurement [4,5]. Predicting plant growth and development will be promising to solve the issues of long cycle, low efficiency, and great uncertainty in the plant breeding industry.

Deep learning models provide the possibility for prediction and simulation of phenotypic traits [6–9]. The effects of environmental factors on plant growth and development were predicted by time-series prediction models. In addition, Yang et al. established

predictive models based on long short-term memory (LSTM) and Convolutional LSTM (ConvLSTM) to predict sunshine hours [10]. Their proposed model could predict the sunshine hours, accumulated precipitation, and average temperature in the next year. At the same time, a data-driven model was developed to predict each growth stage in the work. However, these studies concentrate on prediction and simulation of dynamic changes of a deterministic factor and cannot visualize plant growth and development.

A first attempt was made to predict plant growth and development using historical plant images based on ConvLSTM [5]. Then, the growth change in plant leaves and roots was predicted from time-series images using the generative adversarial network (GAN) [4]. Nevertheless, the images of the growth and development were predicted by these models based on the mask of plants. The texture and color information of plants were ignored during the prediction.

Another generative growth model based on conditional generative adversarial networks was proposed to predict the future appearance of individual plants [11]. A model using the cycle-consistent generative adversarial network (CycleGAN) was proposed to forecast a probable time-series of images with an advancing disease spread [12]. Although realistic looking and reliable images of future plant growth stages were generated by these GAN models, the structural similarity between the generated and real plant images still needs to be improved. Another line of thinking, the prediction model of plant growth and development based on spatiotemporal long short-term memory (ST-LSTM, a variant of the ConvLSTM) was proposed to predict images of future growth and development [13]. The structural similarity between the generated and real plant images was relatively good. However, the timeliness of the prediction by this model was insufficient and the prediction image was blurry. This is because these predictive models discard long-term non-stationary trend information in memory states, resulting in catastrophic forgetting. The processes of plant growth and development can be highly non-stationary in many ways. Examples include low-level non-stationarity such as spatial correlations or temporal dependencies of local pixel values of plant images and high-level variations such as the accumulation and stress adaption of plant growth and development.

The memory in memory network (MIM) was proposed to learn higher-order non-stationarity from spatiotemporal dynamics [14]. MIM uses two cascaded recurrent models to handle the non-stationary and approximately stationary components in the spatiotemporal domain. MIM could compensate for the defects in the lack of ability of LSTM-based variants to model non-stationary trends in the spatiotemporal domain. The above-mentioned studies have been conducted predominantly on the prediction of future growth and development of Brassicaceae, such as Arabidopsis thaliana. The current prediction models still have some problems to improve. Therefore, a prediction model based on ST-LSTM and MIM was proposed in this work.

The major contributions in this work are described as follows.

(1) A novel dataset of wheat growth and development was compiled. The prediction model of plant growth and development was proposed based on ST-LSTM and MIM to predict images of future plant growth.

(2) The performance of the prediction model of plant growth and development was evaluated by calculating SSIM, MSE, and PSNR between the predicted and real plant images. The leaf number, projected area, and length and width of the minimum bounding rectangle of the predicted and real images were also measured and compared to assess whether the prediction of plant growth and development was biologically accurate.

(3) Comparison of evaluation results between the proposed model and the existing models (i.e., a model based only on ConvLSTM [5] and a model based only on ST-LSTM [13]) was conducted.

## 2. Materials and Methods

*2.1. Dataset*

In this work, a novel dataset of wheat growth and development was given, including the multi-view wheat dataset and the successive-view wheat dataset. The multi-view wheat dataset was composed of successive images of four different varieties of wheat: Fielder, Shannong28 (SN28), Jimai22 (JM22), and Kenong199 (KN199). All wheat samples were cultivated in the growth chamber under controlled conditions (23 °C, 16/8 light/dark, 30 Klux, 90% humidity). Every pot of wheat was constantly monitored via an image acquisition platform, which we designed and installed at the growth chamber, as shown in Figure 1a. The image sequences of wheat growth and development in the multi-view wheat dataset were imaged weekly from eight fixed-side views. The obtained sequence for each plant from one view involved eight successive images (Figure 1c). The number of sequences in the dataset was 592. The successive-view wheat dataset composed of successive images of Fielder was also given. The two samples of Fielder were cultivated in the same growth chamber. The images of Fielder samples were taken every half hour from the same view after the two-leaf stage. The obtained sequence for each plant involved 3886 successive images (Figure 1d).

After collecting successive images of wheat growth and development, images were pre-processed to remove the background to avoid any influence of the background on the prediction of growth and development, as shown in Figure 1b. The wheat images in RGB color space were converted into HSV color space. Image segmentation was performed using K-means clustering. A color filter was applied to the HSV images for the extraction of wheat (the green range of images).

The experiment was also conducted on the Panicoid Phenomap-1 dataset [15], which contained 39 varieties of panicoid grain crops. The total group number of panicoid grain crops in the Panicoid Phenomap-1 dataset was 176. The images of panicoid grain crops were captured from 0-degree and 90-degree side views once per day for 29 days. These panicoid grain crops were segmented from the background by auto-thresholding using the Otsu algorithm, and the background was removed by setting it to black. Pre-processed examples of the Panicoid Phenomap-1 dataset content are shown in Figure 1e.

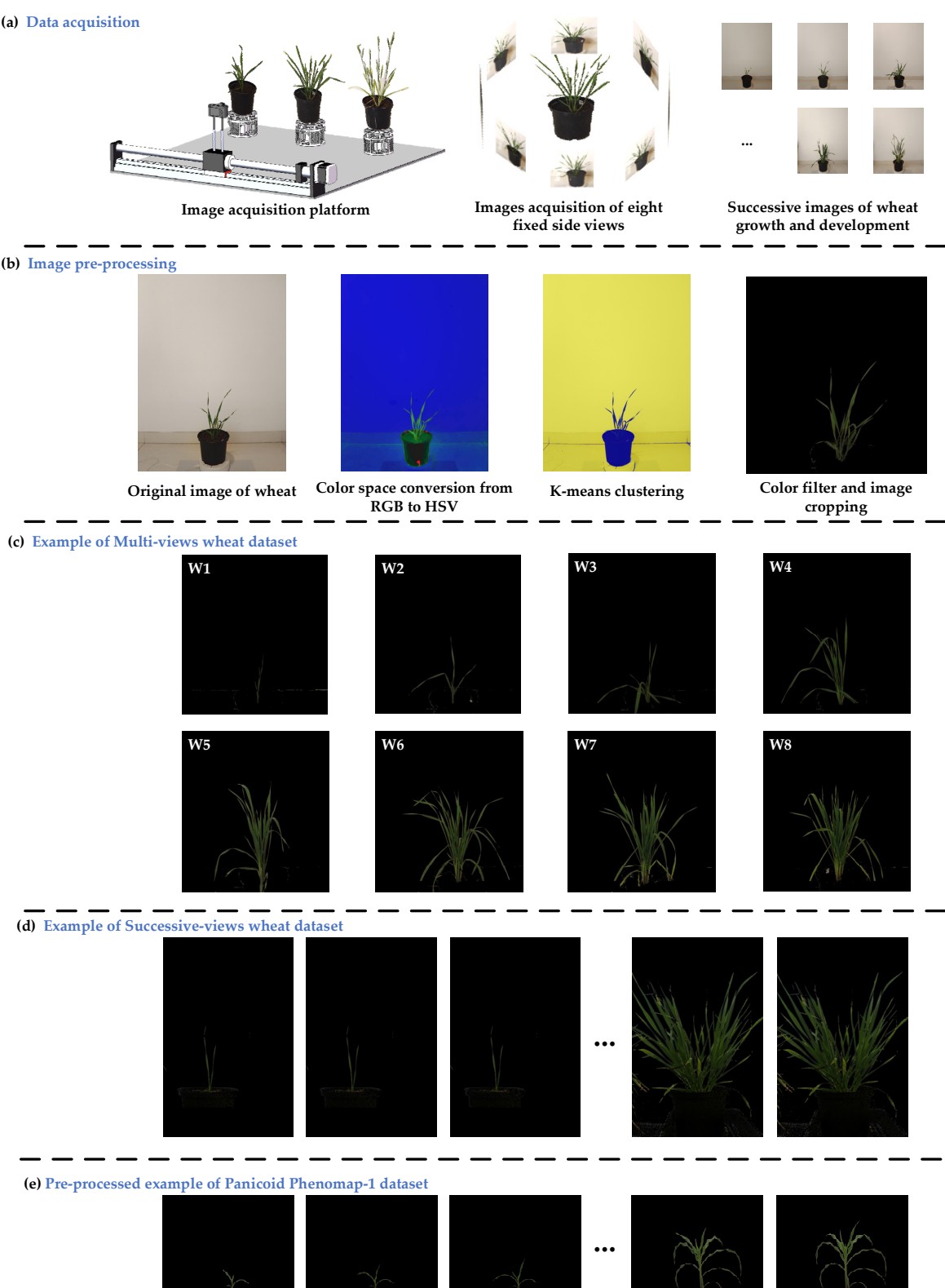

**Figure 1.** Flowchart of dataset construction and examples of the given dataset content. (**a**) Data acquisition using an image acquisition platform we designed and installed at the growth chamber. (**b**) Image pre-processing to obtain plant images without background. (**c**) Examples of the multi-view wheat dataset content. (**d**) Examples of the successive-view wheat dataset content. (**e**) Pre-processed examples of the Panicoid Phenomap-1 dataset content.

### 2.2. Formulation of Plant Growth and Development Predicting Problem

The prediction of plant growth and development could be regarded as a spatiotemporal sequence forecasting problem. The goal of the prediction model is to use the $j$ past images ($I_{t-j+1:t}$) to predict the $k$ future images of plant growth and development ($\hat{I}_{t+1:t+k}$). The prediction process is defined as Equation (1).

$$\hat{I}_{t+1}, \cdots, \hat{I}_{t+k} = \underset{I_{t+1}, \cdots, I_{t+k}}{\operatorname{argmax}} \, p(I_{t+1}, \cdots, I_{t+k} | I_{t-j+1}, \cdots I_t) \tag{1}$$

where $I_t$ is the RGB image of the plant at time $t$ and represented by a tensor $I_t \in \mathbb{R}^{3 \times m \times n}$, $\hat{I}_{t+1}$ is the predicted image of the plant at time $t + 1$.

### 2.3. Prediction Model of Plant Growth and Development

On the tightly coupled spatiotemporal correlation of plant growth and development, a prediction model based on ST-LSTM and MIM was proposed by taking the timing images of plant growth and development as the research object, as shown in Figure 2. The prediction model based on ST-LSTM and MIM could learn stationary variations and the higher-order non-stationarity variations from the dynamics of plant growth and development.

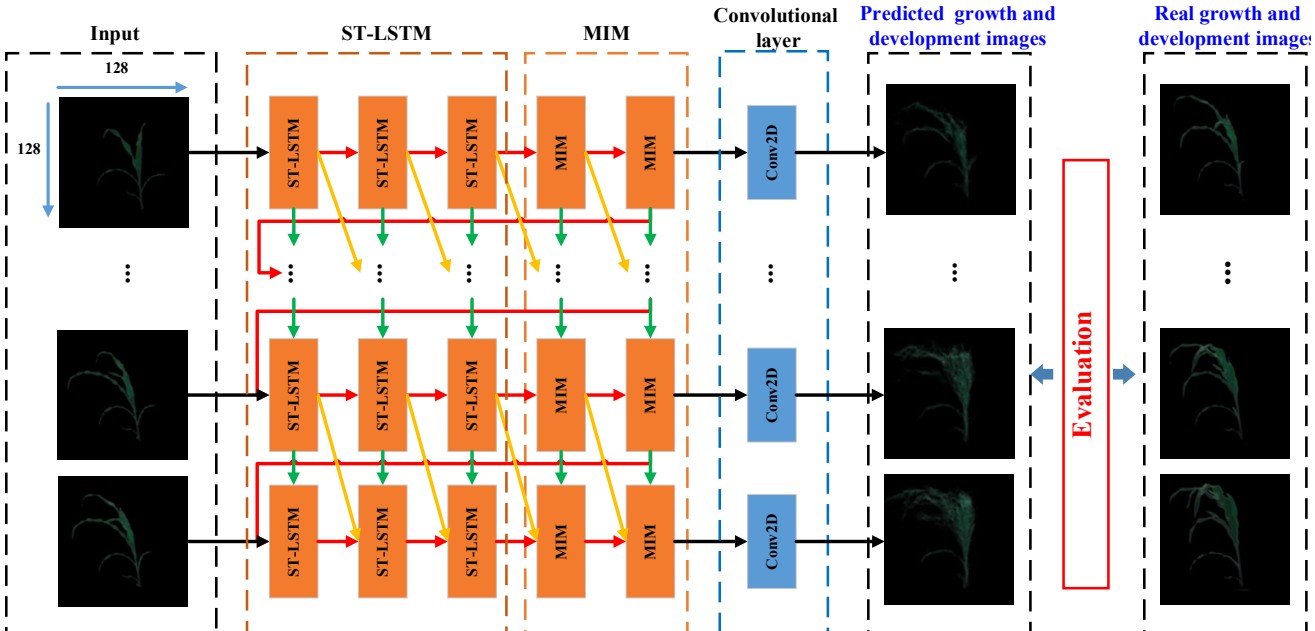

**Figure 2.** Structural diagram of the prediction model for plant growth and development. Yellow arrows: the diagonal state transition paths of $H_{p-1}^{l-1}$ for differential modeling. Green arrows: the horizontal transition paths of the memory cells $C_p^l$, $D_p^l$ and $S_{p-1}^l$. Red arrows: the zigzag state transition paths of $M_p^{l-1}$.

The prediction model consisted of the input layer, the ST-LSTM layer, the MIM layer, the convolutional layer, and the output layer. Three ST-LSTM layers were stacked and each contained 64 ST-LSTM units. Two MIM layers were stacked and each contained 64 MIM units. The input of the prediction model was the $j$ past time-series RGB images of plant growth and development ($I_{t-j+1:t}$), and the output was the predicted $k$ future images of growth and development ($\hat{I}_{t+1:t+k}$). The hidden presentations of spatiotemporal stationarity variations in the time-series images were generated by the ST-LSTM layers. Then, the temporal differencing obtained by subtracting the hidden state $H_{p-1}^{l-1}$ from the hidden state $H_p^{l-1}$ was used as the input of the MIM layers. The temporal features and spatiotemporal features were extracted by the MIM layers. The non-stationarity variations

of the temporal differencing were captured to improve the ability to extract temporal features. As predicted, time-series images of plant growth and development had the same dimensionality as the input, all the temporal and spatiotemporal states were concatenated and fed into a $1 \times 1$ convolutional layer to generate the final prediction.

The ST-LSTM was proposed on the basis of ConvLSTM and introduced a gate-controlled dual memory structure to extract and memorize temporal and spatiotemporal representations simultaneously. A sample of the ST-LSTM unit is shown in Figure 3a. The equations of ST-LSTM are shown as Equations (2)–(11), where '$*$' denotes the convolution operator and '$\circ$' denotes the Hadamard product. The outputs from the ST-LSTM cell were two memory states ($C_p^l$ and $M_p^l$) and a hidden state ($H_p^l$). $C_p^l$ was the temporal memory that was delivered from the $p-1$ node of the same hidden layer. $M_p^l$ was the spatiotemporal memory conveyed vertically from the previous layer at the same time step. These memory states were derived from different directions and concatenated. Different from simple memory concatenation, the ST-LSTM unit used a shared output gate for both memory types to enable seamless memory fusion.

$$g_p = \tanh(\omega_{xg} * X_p + \omega_{hg} * H_{p-1}^l + b_g), \tag{2}$$

$$i_p = \sigma(\omega_{xi} * H_p^{l-1} + \omega_{hi} * H_{p-1}^l + b_i) \tag{3}$$

$$f_p = \sigma(\omega_{xf} * X_p + \omega_{hf} * H_{p-1}^l + b_f) \tag{4}$$

$$C_p^l = f_p \odot C_{p-1}^l + i_p \odot g_p \tag{5}$$

$$g'_p = \tanh(\omega_{xg} * H_p^{l-1} + \omega_{mg} * M_p^{l-1} + b_g) \tag{6}$$

$$i'_p = \sigma(\omega_{xi} * H_p^{l-1} + \omega_{mi} * M_p^{l-1} + b_i) \tag{7}$$

$$f'_p = \sigma(\omega_{xf} * H_p^{l-1} + \omega_{mf} * M_p^{l-1} + b_f) \tag{8}$$

$$M_p^l = f'_t \odot M_p^{l-1} + i'_p \odot g'_p \tag{9}$$

$$o_p = \sigma(\omega_{xo} * X_p + \omega_{ho} * H_{p-1}^l + \omega_{co} * C_p^l + \omega_{mo} * M_p^l + b_o) \tag{10}$$

$$H_p^l = o_p \odot \tanh(\omega_{1 \times 1} * \left[ C_p^l, M_p^l \right]) \tag{11}$$

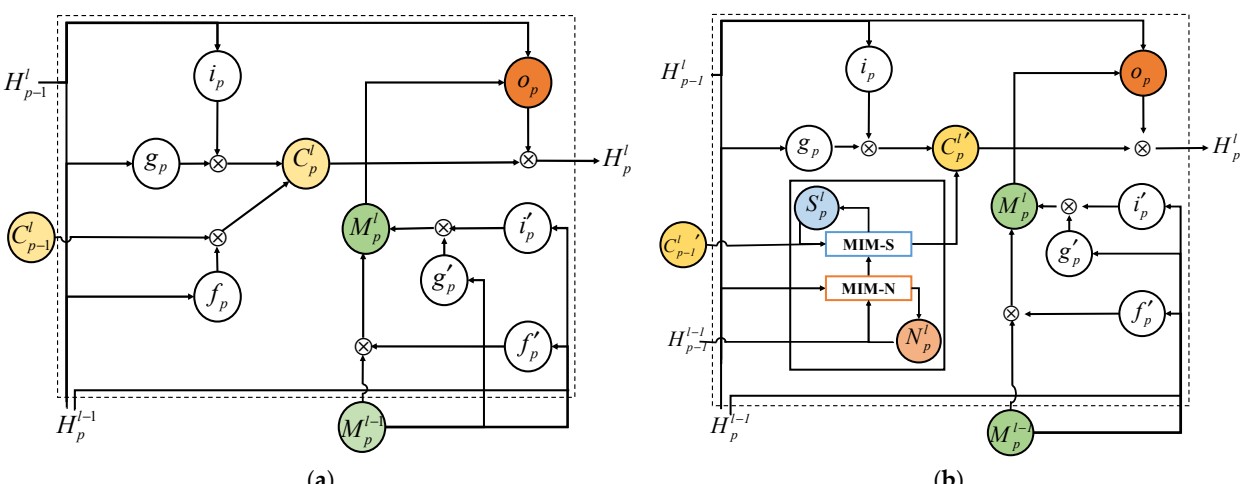

**Figure 3.** Structure diagram of ST-LSTM (**a**) and MIM (**b**) cell.

Based on the idea of difference-stationary assumption, the MIM was proposed to model the non-stationary variations using a series of cascaded memory transitions, as shown in Figure 3b. Two cascaded, self-renewed memory models (non-stationary model

and stationary model) were proposed in the MIM to replace the temporal forget gate $C_p^l$ in the ST-LSTM. Key calculations of the MIM block are shown as Equations (12)–(20). The temporal differencing $(H_p^{l-1} - H_{p-1}^{l-1})$ obtained by subtracting the hidden state $H_{p-1}^{l-1}$ from the hidden state $H_p^{l-1}$ was used as the input of MIM. The new temporal memory $C_{p-1}^{l}{}'$, the spatiotemporal memory $M_p^{l-1}$, and the hidden state $H_{p-1}^l$ were also used as the input of MIM. The non-stationary model (MIM-N) generated the differential features $D_p^l$ to capture the non-stationary variations of the temporal differencing, as shown in Figure 4. The output $D_p^l$ of the MIM-N model and the outer temporal memory $C_{p-1}^{l}{}'$ were taken as inputs of the stationary model (MIM-S) to capture the approximately stationary variations in spatiotemporal sequences, as shown in Figure 4.

$$D_p^l = \text{MIM-N}(H_p^{l-1}, H_{p-1}^{l-1}, N_{p-1}^l), \tag{12}$$

$$T_p^l = \text{MIM-S}(D_p^l, C_{p-1}^{l}{}', S_{p-1}^l), \tag{13}$$

$$C_p^{l}{}' = T_p^l + i_p \odot g_p \tag{14}$$

$$g''_p = \tanh(\omega''_{xg} * (H_p^{l-1} - H_{p-1}^{l-1}) + \omega_{ng} * N_{p-1}^l + b_g), \tag{15}$$

$$i''_p = \sigma(\omega_{xi} * (H_p^{l-1} - H_{p-1}^{l-1}) + \omega_{ni} * N_{p-1}^l + b_i) \tag{16}$$

$$f''_p = \sigma(\omega_{xf} * (H_p^{l-1} - H_{p-1}^{l-1}) + \omega_{nf} * H_{p-1}^l + b_f) \tag{17}$$

$$N_p^l = f''_t \odot N_p^{l-1} + i''_p \odot g''_p \tag{18}$$

$$o''_p = \sigma(\omega_{xo} * (H_p^{l-1} - H_{p-1}^{l-1}) + \omega_{no} * N_p^l + b_o) \tag{19}$$

$$D_p^l = o''_{pt} \odot \tanh(N_p^l) \tag{20}$$

$$g'''_p = \tanh(\omega_{xg} * D_p^{l-1} + \omega_{cg} * C_{p-1}^{l}{}' + b_g), \tag{21}$$

$$i'''_p = \sigma(\omega_{xi} * D_p^{l-1} + \omega_{ci} * C_{p-1}^{l}{}' + b_i) \tag{22}$$

$$f'''_p = \sigma(\omega_{xf} * D_p^{l-1} + \omega_{nf} * C_{p-1}^{l}{}' + b_f) \tag{23}$$

$$S_p^l = f'''_t \odot S_{p-1}^l + i'''_p \odot g'''_p \tag{24}$$

$$o'''_p = \sigma(\omega_{do} * D_p^l + \omega_{co} * C_{p-1}^{l}{}' + \omega_{so} * S_p^l + b_o) \tag{25}$$

$$T_p^l = o'''_{pt} \odot \tanh(S_p^l) \tag{26}$$

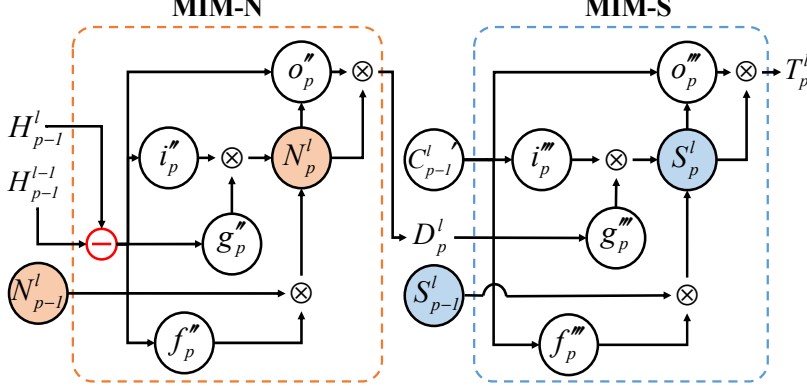

**Figure 4.** Internal structure diagrams of MIM-N and MIM-S. $i''_p$ $g''_p$ $f''_p$ $o''_p$: the input gate, input-modulation gate, forget gate, and output gate in MIM-N. $i'''_p$ $g'''_p$ $f'''_p$ $o'''_p$: the input gate, input-modulation gate, forget gate, and output gate in MIM-S.

*2.4. Evaluation of the Model Performance*

To assess the performance of the prediction model of plant growth and development, the predicted plant images were evaluated quantitatively by using three metrics, including mean square error (MSE) [4], peak signal to noise ratio (PSNR) [16], and structural similarity index measure (SSIM) [17]. MSE calculated by Equation (27) was used to calculate the average squared difference between the predicted image and the real image. A smaller MSE value represents a higher similarity. The PSNR was used to evaluate the level of image noise, as shown in Equation (28). The higher the PSNR value, the better the quality of the predicted image. The SSIM calculated using Equation (29) was used to assess the similarity between the predicted image and the ground truth image. A larger SSIM value represents a higher similarity And SSIM is one where the predicted image is identical to the ground truth image.

$$MSE = \frac{1}{mn} \sum_{i=0}^{m-1} \sum_{j=0}^{n-1} \left[ X_k(i,j) - \hat{X}_k(i,j) \right],\tag{27}$$

$$PSNR = 10 \log_{10} \left( \frac{\max_{X_k}^2}{MSE} \right),\tag{28}$$

$$SSIM = \frac{\left(2\mu_{X_k}\mu_{\hat{X}_k} + c_1\right)\left(2\sigma_{X_k\hat{X}_k} + c_2\right)}{\left(\mu_{X_k}^2 + \mu_{\hat{X}_k}^2 + c_1\right)\left(\sigma_{X_k}^2 + \sigma_{\hat{X}_k}^2 + c_2\right)}\tag{29}$$

Moreover, parameters of the predicted and real images were measured and compared to assess whether the prediction was biologically accurate, such as the leaf number, the projected area, length of the minimum bounding rectangle, and width of the minimum bounding rectangle. The projected area was the pixel number of the plant binary images. The change of projection area could indirectly reflect the growth rate and morphological change of plants. The size of the minimum bounding rectangle could reflect the height and morphological compactness of plants.

## 3. Results and Discussion

The prediction model of plant growth and development was trained on the training and validation dataset, and its performance was evaluated on the testing dataset. The proposed prediction model of plant growth and development consisted of one input layer, three ST-LSTM layers, two MIM layers, one convolutional layer, and one output layer. Three ST-LSTM layers were stacked and each contained 64 ST-LSTM units. Two MIM layers were stacked and each contained 64 MIM units. L2-Loss was chosen as the loss function, and Adam Optimizer with a learning rate of 0.001 was used in the loss function optimization. The batch size was set to two and the number of iterations was set to 80,000 for every training.

In order to validate the performance of the proposed prediction model, it was compared with the existing models, such as the prediction model based only on ConvLSTM [5] and the prediction model based only on ST-LSTM [13]. The comparison of evaluation results between the proposed prediction model and the existing models tested on the given dataset was conducted.

*3.1. Successive-View Wheat Dataset*

In the experimental study, the successive images of Fielder were firstly used to verify the effectiveness of the proposed prediction model. The sliding time window method was used to construct continuous time-series images as the input sequences of the model. The set window length slides along the time axis. The test dataset contained three waves of data (599 successive images at the middle growth stage, 144 successive images at the late growth stage, and 502 successive images at the mid to late growth stage) selected randomly from

successive images of plant growth and development. The remaining successive images were considered for the training and validation dataset.

Five images of future growth and development were predicted at once based on five input images. The window length was set as 10. The prediction model was trained on the training and validation dataset and tested on the test dataset. The qualitative comparison between the predicted images and the real images of future growth and development is shown in Figure 5a. The average values of MSE, PSNR, and SSIM at each step from $t + 1$ to $t + 5$ are shown in Figure 5b. The corresponding parameters (the leaf number, the projected area, and the length and width of the minimum bounding rectangle) of the predicted and real images were measured and compared, as shown in Figure 5c.

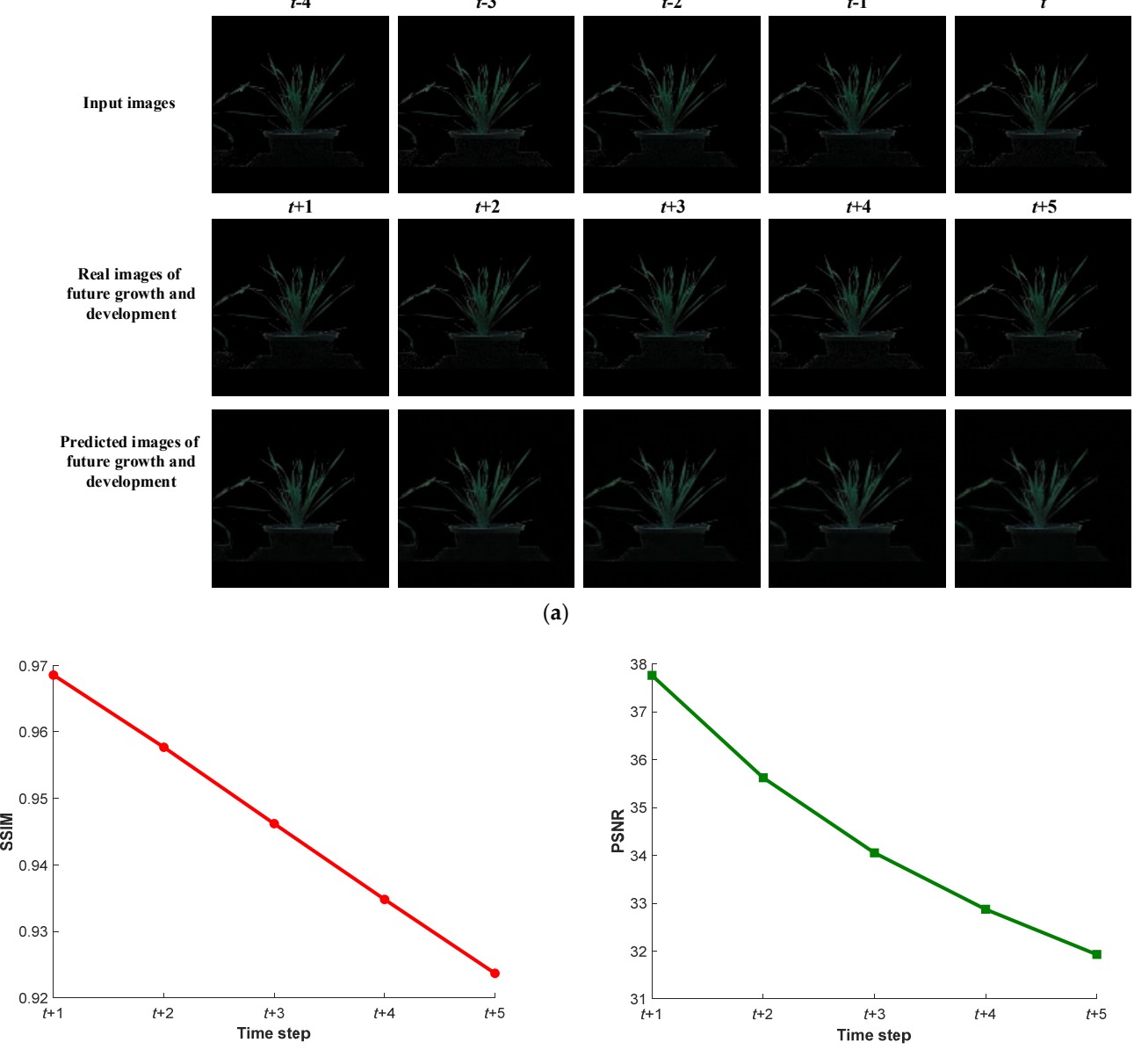

**Figure 5.** *Cont.*

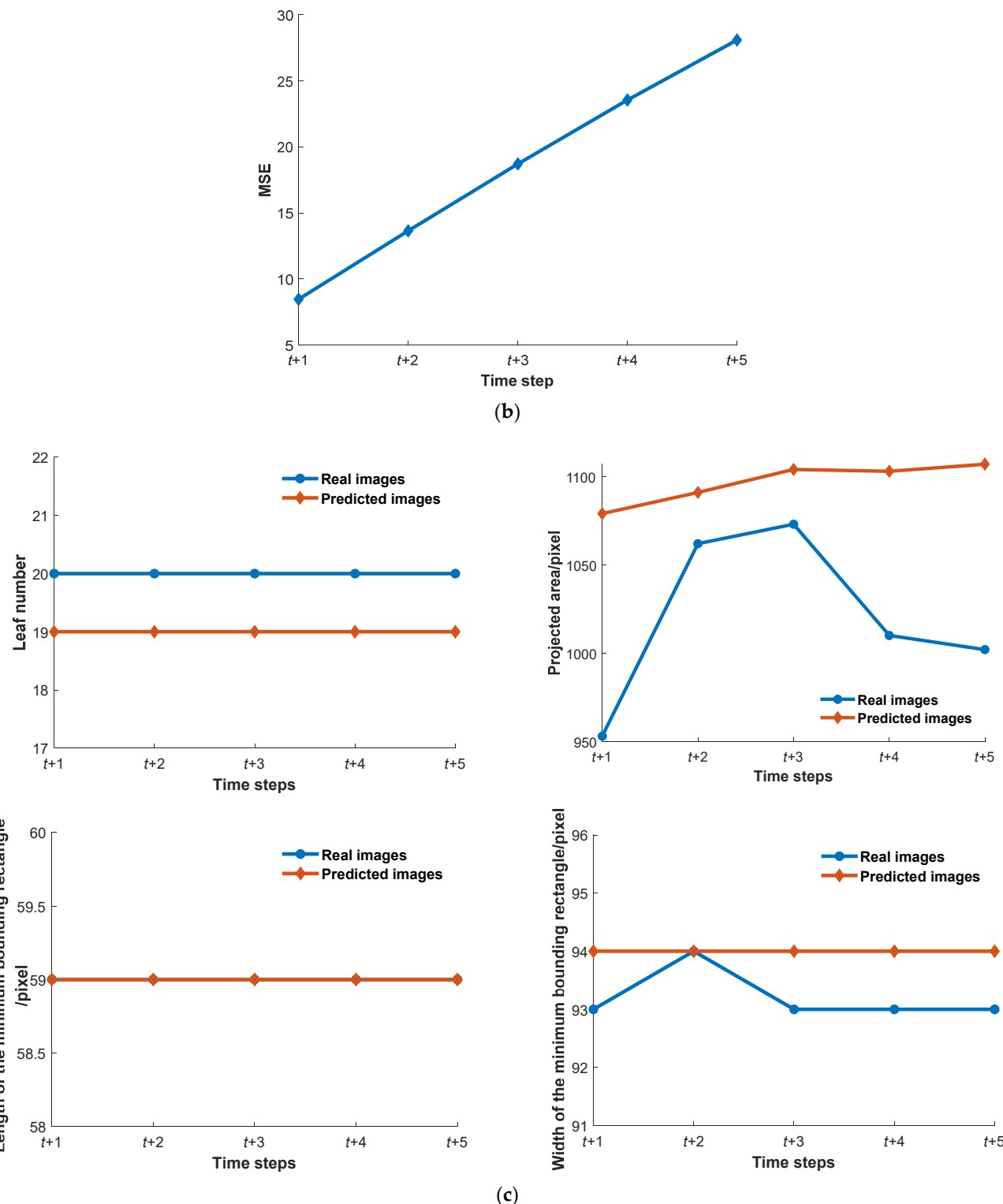

**Figure 5.** Samples of the results predicted by the proposed prediction model on the successive-view wheat dataset. (**a**) Comparison between the predicted images and the real images of future growth and development. (**b**) Average evaluation results of MSE, PSNR, and SSIM at each time step from *t* + 1 to *t* + 5. (**c**) Comparison of leaf number, projected area, length of the minimum bounding rectangle, and width of the minimum bounding rectangle between the predicted and real images.

The SSIM values surpassed 91% for all time steps. The mean of MSE values was 18.50 and the MSE values were below 30 for all time steps. The mean of PSNR values was 34.45. The smaller the value of MSE, the better the predictive ability of the proposed model. The results showed a high degree of similarity between the predicted images and the real images of plant growth and development. The values of PSNR and SSIM typically showed a gradual decrease trend as time step increased and the values of MSE showed a gradually increasing trend as time step increased. This may be caused by deviations of the predictions accumulated over time and the complexity of plant growth and development. Because the time interval between the two steps was only 30 min, the leaf number, and the length and width of the minimum bounding rectangle of real images had fewer changes and were similar to the ones of the predicted images, as shown in Figure 5c. Yet, the rhythmic leaf movement and the growth of young leaves may change the projected area. There was a larger discrepancy between the projected area of predicted images and the projected area of real images. The changing trend of the projected area of predicted images is similar to that of real images. These results reflected the predictive validity of the proposed model.

The prediction of plant growth and development at different numbers of time steps and different time intervals between two steps were compared to determine the optimal number of time steps and optimal time interval between two steps. The spacer input sequences of the model were acquired from the dataset at the set number of prediction steps and the time interval between time steps. The prediction results of different numbers of steps and different time intervals between steps are shown in Figure 6.

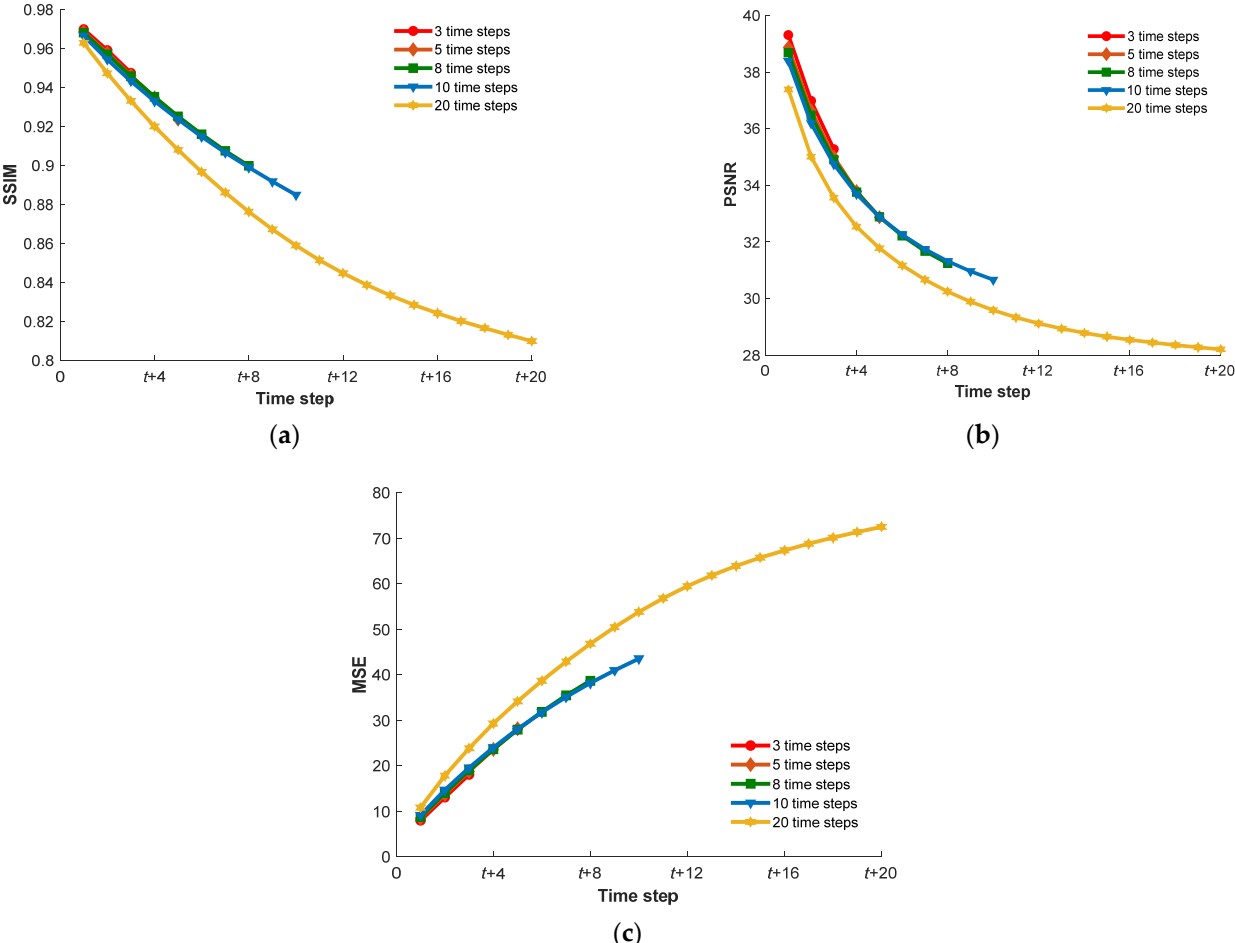

**Figure 6.** Average evaluation results of MSE (**a**), PSNR (**b**), and SSIM (**c**) at each time step where the number of prediction steps was set as 3, 5, 8, 10, and 20.

The values of MSE, PSNR, and SSIM at each step between the predicted images with the real images are shown in Figure 6, where the time interval between two steps was set as 30 min and the number of prediction steps was set as 3, 5, 8, 10, and 20. When the number of prediction steps was 20, the values of PSNR, and SSIM at each step were significantly lower than other results and the values of MSE at each step were significantly higher than other results. When the number of prediction steps was lower than 10, the values of MSE at each step (from $t + 1$ to $t + 5$) increased as the number of prediction steps increased (Figure 6c), and the values of PSNR and SSIM at each step (from $t + 1$ to $t + 5$) were increased as the number of prediction steps increased (Figure 6a,b). The standard deviations of MSE and PSNR values at each step were both less than two. The standard deviation of SSIM values at each step was smaller than 0.01. These results illustrate that the number of prediction steps had a comparatively small effect on the performance of the proposed prediction model until it was greater than 10.

Next, the performances of the proposed prediction model tested on different time intervals were evaluated to explore the effect of the time interval between two steps on the proposed prediction model. The average values of MSE, PSNR, and SSIM at each step from $t + 1$ to $t + 5$ are shown in Figure 7, where the time interval between two steps was set as 30 min, 1 h, 2 h, 6 h, and 12 h. The values of MSE at each step (from $t + 1$ to $t + 5$) were increased as the time interval between the two steps increased (Figure 7c), and the values of PSNR and SSIM at each step (from $t + 1$ to $t + 5$) were increased as the number of prediction steps increased (Figure 7a,b). However, when the time interval between the two steps was set as 2 h, the values of MSE at each step were significantly increased. When the time interval between the two steps was set as 6 h, the values of MSE at each step obtained were close to those obtained by setting the time intervals as 12 h. When the time interval between two steps was set as 6 h, the SSIM values surpassed 73% for all time steps and the SSIM value at the first time step was 79.85%, the mean of PSNR values was 26.68 and the mean of MSE values was 34.45. These results illustrate that the time interval had an extremely large effect on the performance of the proposed prediction model. In order to achieve 85% SSIM between the prediction and real plant images, the time interval needs to be set to 1 h. Therefore, for a more reliable and longer-term prediction of plant growth and development, the optimal number of time steps is 10 and the optimal time interval between two steps is 1 h.

Under the optimal setting, the performance of the proposed prediction model and the existing models are shown in Figure 8. The mean of PSNR values of the proposed prediction model was 30.67. The SSIM values of the proposed prediction model surpassed 85% for all time steps, which was higher than the ones of the prediction model based only on ConvLSTM and the prediction model based only on ST-LSTM. The mean of MSE values of the proposed prediction model was 46.11 and the MSE values of the proposed prediction model were below 68 for all time steps. The proposed prediction model was not good at the MSE. The MSE values of the proposed prediction model at each step (from $t + 4$ to $t + 10$) were higher than the ones of the prediction model based only on ConvLSTM and the prediction model based only on ST-LSTM. As shown in Figure 5c, the projection area of wheat was as high as 1000 and the projection area was also the pixel number of the binary images. The relative difference of MSE between the proposed prediction model and the existing models was less than 30, which was acceptable. The results above validated the proposed prediction model and showed its robustness as compared with the existing models.

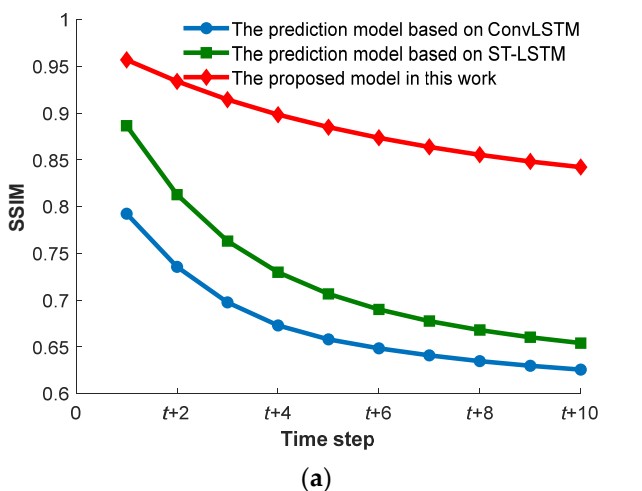

**Figure 7.** Average evaluation results of MSE (**a**), PSNR (**b**), and SSIM (**c**) at each time step where the time interval between two steps was set as 30 min, 1 h, 2 h, 6 h, and 12 h.

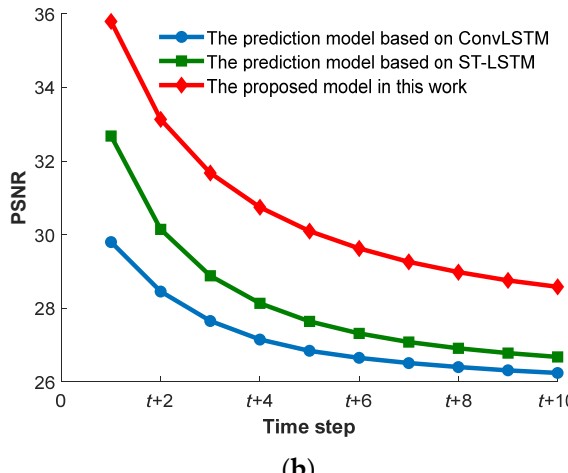

**Figure 8.** *Cont.*

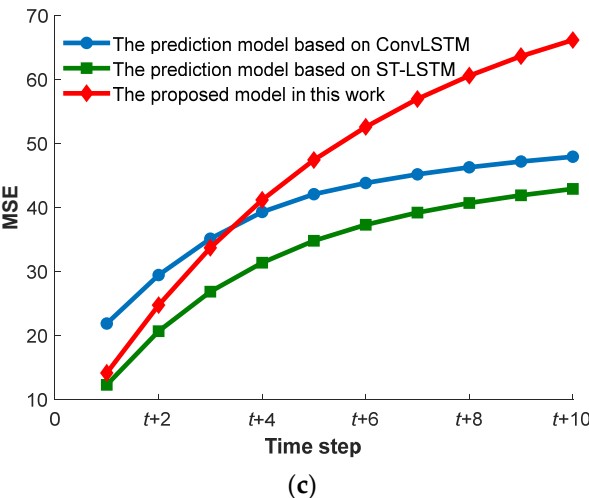

(**c**)

**Figure 8.** Comparison of MSE (**a**), PSNR (**b**), and SSIM (**c**) between the proposed prediction model and the existing models, where the time intervals between two steps were set as 1 h.

### 3.2. Multi-View Wheat Dataset

In the experimental study, the successive images of four different varieties of wheat without background were used to further verify the effectiveness of the proposed model in predicting the growth and development of different varieties and different views. The test dataset contained 144 sequences (obtained from eight views of 18 plants) selected randomly from the multi-view wheat dataset. The remaining sequences of successive images were considered for the training and validation dataset. The prediction model of plant growth and development was trained on the training and validation dataset. Two images of future growth and development were predicted at once based on three input images.

The proposed prediction model of growth and development was tested on the test dataset. The qualitative comparison between the predicted images and the real images of future growth and development is shown in Figure 9. The average values of MSE, PSNR, and SSIM at each step from $t + 1$ to $t + 2$ were also calculated at different time steps to evaluate the predicted result of wheat growth and development. The values of SSIM at $t + 1$ and $t + 2$ steps were 81.63% and 80.46%. These results illustrate the validity of the proposed model again. When the time interval was increased, the predicted plant growth and development images could still have relatively good structural similarity with the real images by increasing the amount of training data.

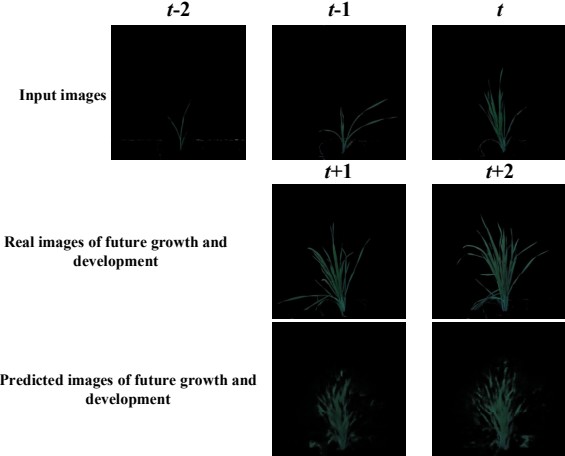

**Figure 9.** Comparison between the predicted images and the real images of future growth and development of the experimental results on the multi-view wheat dataset. The past 3 images ($I_{t-2:t}$) were used to predict the future images of plant growth and development ($\hat{I}_{t+1:t+2}$). $I_t$ was the RGB image of the plant at time $t$.

### 3.3. Panicoid Phenomap-1 Dataset

The proposed prediction model was also evaluated on the Panicoid Phenomap-1 dataset. Successive images of 39 varieties of panicoid grain crops without background were used to verify the effectiveness and robustness of the proposed model in predicting growth and development. The test dataset contained 39 randomly chosen groups containing all genotypes of panicoid grain crops. The remaining 137 sequences of successive images were considered for the training and validation dataset. The prediction model was trained on the training and validation dataset and tested on the test dataset. Five images of future growth and development were predicted at once based on five input images. The window length was set as 10. The predicted images and the real images of future growth and development are shown in Figure 10. The corresponding parameters (leaf number, projected area, length and width of the minimum bounding rectangle) of the predicted and real images were measured and compared, as shown in Figure 11. The average values of MSE, PSNR, and SSIM at each step from $t + 1$ to $t + 5$ are also shown in Figure 12.

The predicted results obtained by the proposed model on the Panicoid Phenomap-1 dataset were similar to the results above. The leaf number, projected area, and length and width of the minimum bounding rectangle of the predicted images were comparable and showed good agreement with the ones of real images. However, accuracies for late prediction time steps were lower, especially for the length of the minimum bounding rectangle. This problem is also reflected in the changing trend of the SSIM, MSE, and PSNR. When the number of time steps was set as five, the values of PSNR and SSIM typically decreased and the values of MSE increased with time. This again validated that the predicted results gradually become worse and may be caused by deviations of the predictions accumulated over time and the complexity of plant growth and development.

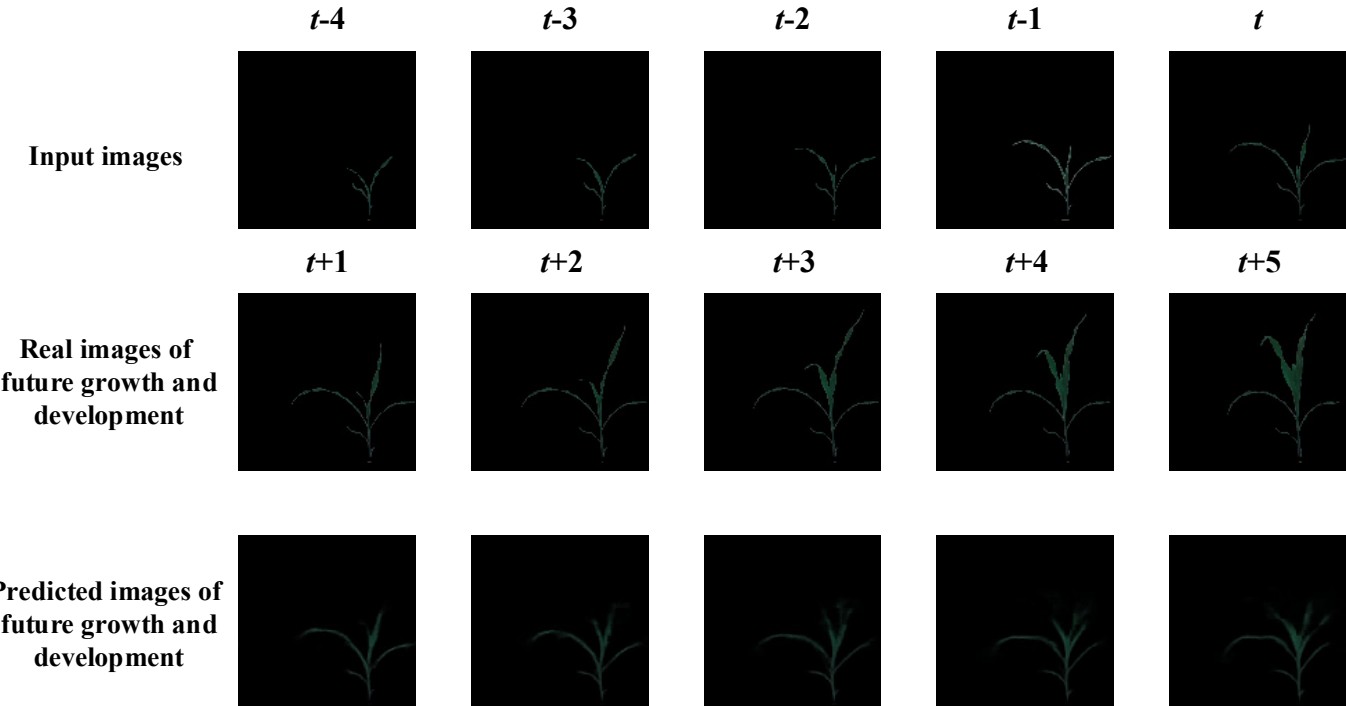

**Figure 10.** Comparison between the predicted images of panicoid grain crops and the real images of future growth and development of the experimental results on the Panicoid Phenomap-1 dataset.

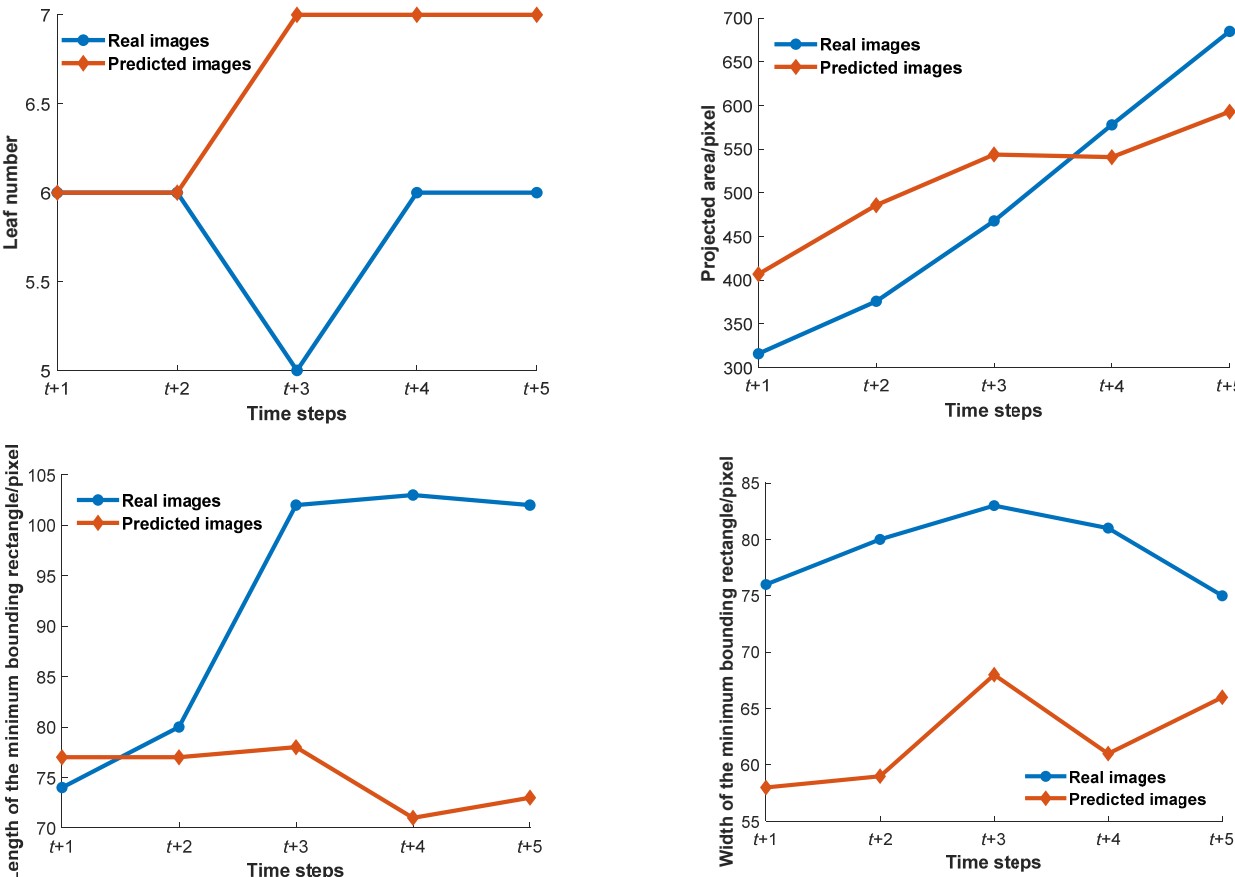

**Figure 11.** Comparison of leaf number, projected area, and length and width of the minimum bounding rectangle between the predicted and real images of the experimental results on the Panicoid Phenomap-1 dataset.

On the other hand, the number of prediction steps was set as 3, 5, 8, 10, and 20. The average values of MSE, PSNR, and SSIM at each step were also calculated to determine the optimal number of prediction steps, as shown in Figure 12. The values of MSE at each step were first decreased and then increased as the number of prediction steps increased (Figure 12c), and the values of PSNR and SSIM at each step were first increased and then decreased as the number of prediction steps increased (Figure 12a,b). The standard deviations of PSNR values at each step were less than two and the standard deviations of SSIM values at each step were smaller than 0.02. These results again illustrated that the number of prediction steps had a comparatively small effect on the performance of the proposed prediction model. However, a larger difference was found regarding the MSE values at each step obtained by setting different numbers of prediction steps. When the number of prediction steps was set to eight, the model had the best prediction performance on the Panicoid Phenomap-1 dataset. The SSIM values surpassed 78% for all time steps. The mean of MSE values was 77.78 and the MSE values were below 118 for all time steps. The mean of PSNR values was 29.03. The trend of the predicted results on the Panicoid Phenomap-1 dataset was different from that of the successive-view wheat dataset. This may be caused by the increase in the time intervals. The time intervals between two steps of the successive-view wheat dataset were less than 12 h. The real images of plant growth and development at the time step $t + 1$ bore a strong visual resemblance to the real images at the time step $t + 1$. However, the time interval of the Panicoid Phenomap-1 dataset was 24 h. With the number of prediction steps increased, the proposed model can better model the dynamic of plant growth and development to predict plant future growth and development. In parallel, deviations of the predictions accumulated over time became

more conspicuous. Therefore, when the number of prediction steps was set to eight, the model had the best prediction performance on the Panicoid Phenomap-1 dataset.

Under the optimal setting, the performances of the proposed prediction model and the existing models are shown in Figure 13. The SSIM and PSNR values of the proposed model are significantly higher than those of the prediction model based only on ConvLSTM and the prediction model based only on ST-LSTM. The proposed prediction model was also not good at the MSE. The results were similar to the results above. The MSE values of the proposed prediction model at each step were higher than the ones of the prediction model based only on ConvLSTM and the prediction model based only on ST-LSTM. The relative difference of MSE between the proposed prediction model and the existing models was less than 40. These results above again validated the robustness of the proposed prediction model.

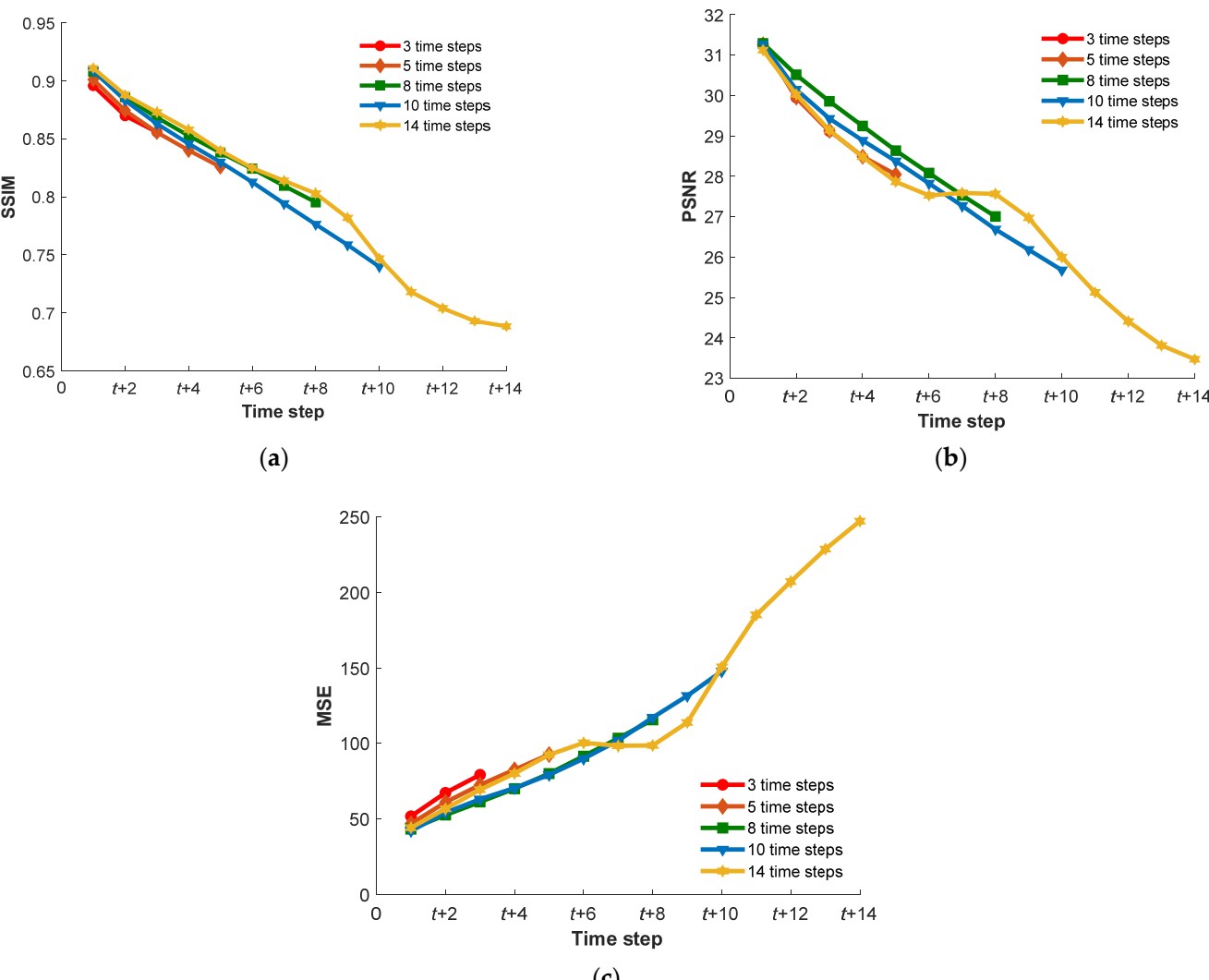

**Figure 12.** Average evaluation results of MSE (**a**), PSNR (**b**), and SSIM (**c**) at each time step on the Panicoid Phenomap-1 dataset, where the number of prediction steps was set as 3, 5, 8, 10, and 20.

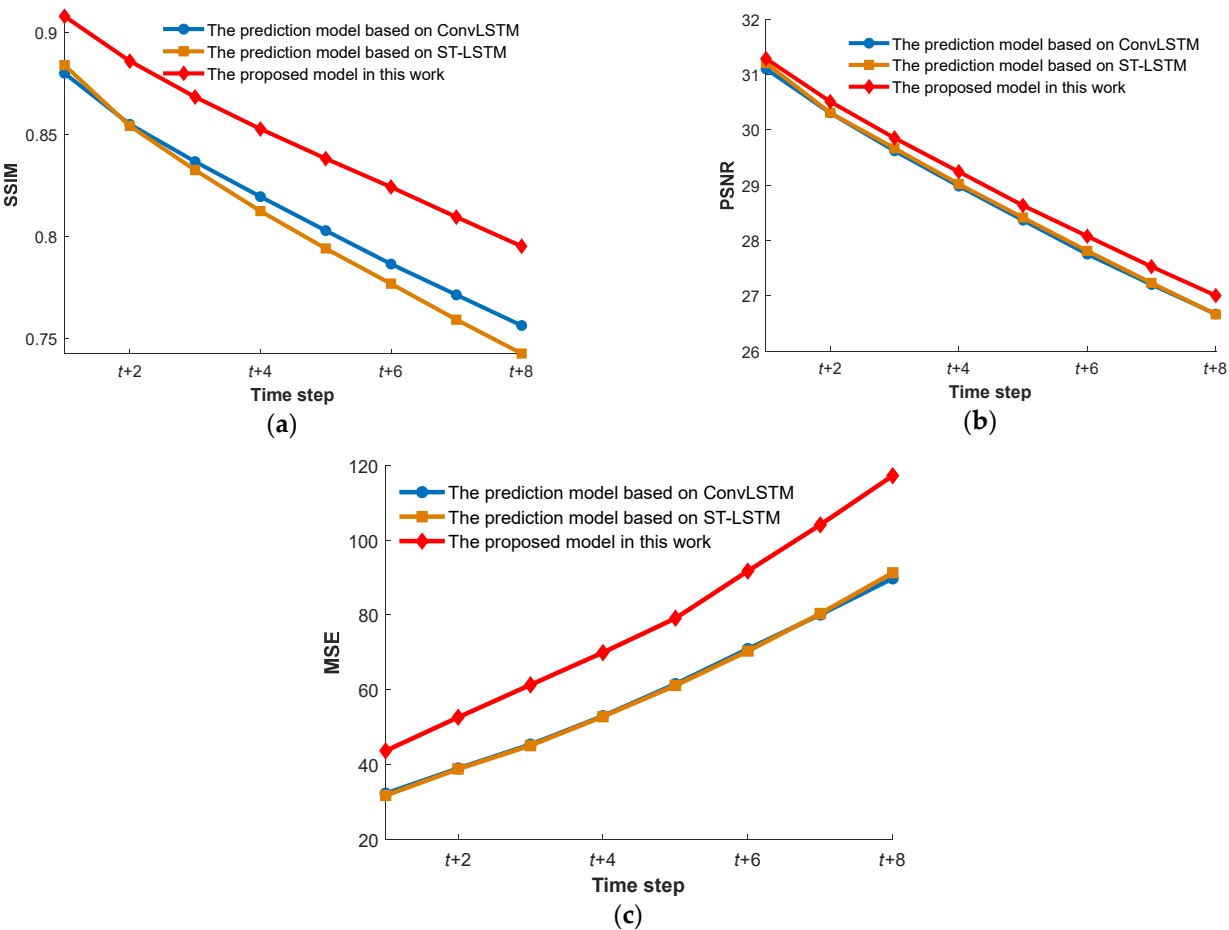

**Figure 13.** Comparison of evaluation results (MSE (**a**), PSNR (**b**), and SSIM (**c**)) between the proposed prediction model and the existing models tested on the Panicoid Phenomap-1 dataset.

Compared with others' work using GAN models, the proposed model did not perform well in terms of the blurriness of the predicted images. Nevertheless, there are two advantages of this work. The first advantage is that the images of future plant growth and development predicted by the proposed model have higher structural similarities with the real images. The proposed model predicted the future images of plant growth and development by modeling the dynamic behaviors of plant growth and development using ST-LSTM and MIM modules. The hidden presentations of spatiotemporal stationarity variations in the time-series images were generated by the ST-LSTM layers. The MIM exploits the differential signals between adjacent recurrent states to model the non-stationary and approximately stationary properties in spatiotemporal dynamics with two cascaded, self-renewed memory models. By stacking multiple MIMs, we could potentially handle higher-order non-stationarity of plant growth and development. The second advantage is that the effect of the number of time steps and the optimal time interval between two steps on the prediction performance of the proposed model was analyzed and the optimal number of time steps and the optimal time interval between two steps were determined, which will provide a valuable reference for the prediction studies of plant growth and development.

## 4. Conclusions

In this work, we proposed a prediction model of plant growth and development to reliably generate images of future plant growth stages and give a novel dataset of wheat growth and development. The performance of the prediction model of plant growth and development was evaluated by calculating SSIM, MSE, and PSNR between the predicted

and real plant images. The leaf number, projected area, and length and width of the minimum bounding rectangle of the predicted and real images were also measured and compared to assess whether the prediction was biologically accurate. Findings reveal a high degree of consistency and similarity between the predicted and real plant frames. Moreover, the optimal number of time steps and the optimal time interval between two steps were determined to provide a valuable reference for the prediction studies of plant growth and development. The comparison of evaluation results between the proposed prediction model and the existing models tested on the given dataset was conducted to validate its robustness. The proposed model also could be retrained and adapted to other domains, such as the prediction of plant growth and development on the effects of abiotic and biotic stresses. This work could potentially speed up biologists', geneticists', and breeders' experimental cycles by reducing the time required to grow, image, and measure plants and further accelerate breeding for addressing the challenge of declining food security.

**Author Contributions:** Conceptualization, C.W., P.L. and X.L.; methodology, C.W.; software, C.W.; validation, C.W., W.P. and X.S.; formal analysis, C.W.; investigation, C.W., H.Y. and J.Z.; resources, C.W., H.Y., P.L., J.Z. and X.L.; data curation, C.W. and W.P.; writing—original draft preparation, C.W.; writing—review and editing, C.W., P.L. and X.S.; visualization, C.W. and W.P.; supervision, X.L.; project administration, P.L. and X.L.; funding acquisition, P.L. and X.L. All authors have read and agreed to the published version of the manuscript.

**Funding:** This research was funded by the Natural Science Foundation of Shandong Province (ZR2020KF002); Shandong Provincial Key Research and Development Plan (Major Science and Technology Innovation Project) (2021LZGC013; 2021TZXD001); and NSFC (31871543). The authors are grateful to all study participants.

**Data Availability Statement:** The data presented in this work are available on request from the corresponding author.

**Acknowledgments:** We would like to acknowledge the State Key Laboratory of Crop Biology, College of Mechanical and Electronic Engineering, Shandong Agricultural University, and Shandong Provincial Key Laboratory of Horticultural Machinery and Equipment for infrastructural support.

**Conflicts of Interest:** The authors declare no conflict of interest.

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
