# Peer review of "Predicting Plant Growth and Development Using Time-Series Images"

_agronomy, doi:10.3390/agronomy12092213_

Round 1

Reviewer 1 Report

The paper lags in novelty.

The results should be compared with the existing methods.

Author Response

Thanks for your suggestion. The evaluation results of the proposed model had been compared with the ones of the existing methods (ConvLSTM, ST-LSTM). The SSIM and PSNR values of the proposed prediction model were significantly higher than the ones of the the existing methods. The comparison of evaluation results had been added in Section 3 “Results and discussion”. The proposed prediction model of plant growth and development could potentially speed up biologists, geneticists, and breeders’ experimental cycles by reducing the time required to grow, image, and measure plants and further accelerate breeding for addressing the challenge of declining food security. Therefore, this work was highly innovative. All revisions were highlighted in the revised manuscript. 

Reviewer 2 Report

The research objective was to predict plant growth and development with a modified deep learning model. Some image combinations were used for the model input, and the future images were predicted. The structure of the model could have some novelty, but the experiment design seems not to be appropriate. The manuscript must be significantly revised because of the following reasons:

1.     Plant images are not plant growth and development. The images do not accord with the plant height, dry weight, number of leaves, leaf area, etc. We do not regard image prediction as growth prediction, although we can infer growth from the appropriate images. This is the most crucial problem. We do not always need the plant images to know the plant growth and development. It should be distinguished.

2.     The methods are not appropriately explained. For example, some essential values, such as learning rate, batch size, number of layers, and so on, were omitted.

3.     The experiment design has to be improved. For example, the authors only changed the input image settings. However, other variation, such as output type, models, and so on, have to be compared for the model robustness.

Author Response

Question 1: Plant images are not plant growth and development. The images do not accord with the plant height, dry weight, number of leaves, leaf area, etc. We do not regard image prediction as growth prediction, although we can infer growth from the appropriate images. This is the most crucial problem. We do not always need the plant images to know the plant growth and development. It should be distinguished.Reply: Thanks for your suggestion. We identified that the prediction model of plant growth and development in this work was proposed to predict the image sequences of future growth and development. And we revised the expression and highlighted it in the revised manuscript. What’s more, the leaf number, the projected area, length of the minimum bounding rectangle, and width of the minimum bounding rectangle of the predicted and real images had been added in Section 3 “Results and discussion” to reflect the growth and development.Question 2: The methods are not appropriately explained. For example, some essential values, such as learning rate, batch size, number of layers, and so on, were omitted.Reply: Thanks for your suggestion. The learning rate, batch size, and the number of layers had been added in Section 3 “Results and discussion” and highlighted in the revised manuscript.  Question 3: The experiment design has to be improved. For example, the authors only changed the input image settings. However, other variation, such as output type, models, and so on, have to be compared for the model robustness.Reply: Thanks for your suggestion. The experiment design had been improved in Section 3 “Results and discussion”. The leaf number, the projected area, length of the minimum bounding rectangle, and width of the minimum bounding rectangle parameters of the predicted and real images had been added to assess whether the prediction of plant growth and development was biologically accurate. Comparison of evaluation results between the proposed prediction model and the existing models (ConvLSTM, ST-LSTM) tested on the given dataset was added to validate model robustness. All revisions were highlighted in the revised manuscript.

Reviewer 3 Report

Authors presented a good idea of "Predicting Plant Growth and Development Using Time-series 3 Images". Here, authors have also considered night images, that is a good step. But some point of improvements are still there, like:

The comparison should be with latest techniques. So authors are encouraged to do with atleast 1 like "A Deep Learning-Based Novel Approach for Weed Growth Estimation".

The contribution points needs to be mentioned in the end of Introduction section. This will improve the readability of the work. 

Can include some latest work like a) A hybrid convolutional neural network model for diagnosis of COVID-19 using chest X-ray images b) Intelligent Method for Detection of Coronary Artery Disease with Ensemble Approach

Author Response

Question 1: The comparison should be with latest techniques. So authors are encouraged to do with atleast 1 like "A Deep Learning-Based Novel Approach for Weed Growth Estimation".Reply: Thanks for your suggestion. The comparison of evaluation results between the proposed prediction model and the existing models (ConvLSTM, ST-LSTM) tested on the given dataset had been added in Section 3 “Results and discussion” and highlighted in the revised manuscriptQuestion 2: The contribution points needs to be mentioned in the end of Introduction section. This will improve the readability of the work.Reply: Thanks for your suggestion. The contribution points had been added at the end of the Introduction sectionQuestion 3: Can include some latest work like a) A hybrid convolutional neural network model for diagnosis of COVID-19 using chest X-ray images b) Intelligent Method for Detection of Coronary Artery Disease with Ensemble ApproachReply: Thanks for your suggestion. Some latest work had been cited and highlighted in the Introduction section.

Round 2

Reviewer 2 Report

The manuscript was adequately editted.

Reviewer 3 Report

Seems Acceptable now.